# Processes of care and survival associated with treatment in specialist teenage and young adult cancer centres: results from the BRIGHTLIGHT cohort study

Lorna A Fern,[1] Rachel M Taylor [ID] ,[2] Julie Barber,[3] Javier Alvarez-Galvez [ID] ,[4] Richard Feltbower,[5] Sarah Lea,[6] Ana Martins,[7] Stephen Morris [ID] ,[8] Louise Hooker,[9] Faith Gibson,[10,11] Rosalind Raine,[12] Dan P Stark,[13] Jeremy Whelan [ID] [6]

For numbered affiliations see end of article.

**Correspondence to**
Dr Rachel M Taylor;
rtaylor13@nhs.net

## ABSTRACT

**Objective** Survival gains in teenagers and young adults (TYA) are reported to be lower than children and adults for some cancers. Place of care is implicated, influencing access to specialist TYA professionals and research. Consequently, age-appropriate specialist cancer care is advocated for TYA although systematic investigation of associated outcomes is lacking. In England, age-appropriate care is delivered through 13 Principal Treatment Centres (TYA-PTC). BRIGHTLIGHT is the national evaluation of TYA cancer services to examine outcomes associated with differing places and levels of care. We aimed to examine the association between exposure to TYA-PTC care, survival and documentation of clinical processes of care.

**Design** Prospective cohort study.

**Setting** 109 National Health Service (NHS) hospitals across England.

**Participants** 1114 TYA, aged 13–24, newly diagnosed with cancer between 2012 and 2014.

**Intervention** Participants were assigned a TYA-PTC category dependent on the proportion of care delivered in a TYA-PTC in the first year after diagnosis: all care in a TYA-PTC (ALL-TYA-PTC, n=270), no care in a TYA-PTC (NO-TYA-PTC, n=359), and some care in a TYA-PTC with additional care in a children's/adult unit (SOME-TYA-PTC, n=419).

**Primary outcome** Data were collected on documented processes indicative of age-appropriate care using clinical report forms, and survival through linkage to NHS databases.

**Results** TYA receiving NO-TYA-PTC care were less likely to have documentation of molecular diagnosis, be reviewed by a children's or TYA multidisciplinary team, be assessed by supportive care services or have a fertility discussion. There was no significant difference in survival according to category of care. There was weak evidence that the association between care category and survival differed by age (p=0.08) with higher HRs for those over 19 receiving ALL or SOME-TYA-PTC compared with NO-TYA-PTC.

**Conclusion** TYA-PTC care was associated with better documentation of clinical processes associated with age-appropriate care but not improved survival.

## Strengths and limitations of this study

► Our study is the first prospective longitudinal national evaluation of specialist cancer care for teenagers and young adults with cancer.

► Using routinely collected National Health Service (NHS) data, we were able to assign participants into three groups according to how much exposure to specialist care they had received in the first year following diagnosis.

► Multiple data sources from patients, NHS and clinical report forms allowed us to adjust for multiple predefined confounding variables.

► Specialist services for teenagers and young adults have evolved since recruitment and may not reflect current service configuration.

► Our study did not meet its anticipated recruitment target, recruiting 20% of the total population diagnosed during the recruitment period and this may limit generalisability of the results.

## INTRODUCTION

Cancer in teenagers and young adults (TYA) aged 15–24 years is rare, comprising approximately 1% of the total cancer population in the UK.[1] Historically when compared with children and older adults, TYA with cancer have experienced lower survival improvements for certain cancers. Prolonged pathways to cancer diagnosis, less research, an inadequate understanding of cancer biology in young people, poor choice of treatment protocols and place of care are all implicated.[2–7] It is now generally accepted that neither children's nor adult cancer services may fully meet the needs of young people with cancer who characteristically present with a spectrum of rare cancer types requiring specialist site specific expertise and additional psychological, educational and social support.[8–11]

TYA cancer care is increasingly recognised as an essential specialism. In England, the

National Institute for Health and Clinical Excellence (NICE) published *Improving Outcome Guidance for Children and Young People with Cancer* in 2005, which provided details on how care should be delivered to TYA.[12] Central to this guidance was the establishment of specialist TYA Principal Treatment Centres (TYA-PTC) and a mandate that young people aged 15–18 years *must* receive care in a TYA-PTC and those aged 19–24 years should have *unhindered access* to age-appropriate care but could chose to have care more locally to home in an adult cancer unit. Where care is delivered influences clinical outcomes and centralisation of care for rare cancers is advocated. Treatment of cancer in children in a limited number of UK centres since the 1960s contributed to improvements in survival.[13 14] For young people, place of care will influence access to clinical trials, treatment protocol (paediatric vs adult regimens) and access to a wider team specialising in TYA care, all of which could influence outcomes. In England, young people have free access to healthcare and can receive all of their care in a TYA-PTC, or all of their care in a children's or adult cancer unit, or they can receive care between these institutions having some care delivered in the TYA-PTC with additional components being delivered by children's/adults services. The decision-making behind referral into a TYA-PTC or a children's/adult unit is not fully understood and is likely to be influenced by local pathways and by older TYA being offered a choice.

Despite a lack of evidence, it has been assumed by professionals and young people themselves that age-appropriate care delivered in a specialist environment will positively impact outcomes. Age-appropriate services usually include access to a specialist environment, referral to specialist TYA multidisciplinary team (MDT) meetings in addition to a cancer-site specific MDT, consideration of clinical trial accrual, provision of age-appropriate information, opportunities to preserve fertility and referral to support services such as a TYA clinical nurse specialist, psychologist, social worker or youth support coordinator. The components of this service that influence outcomes are not fully described, although increasingly specialist TYA metrics and potential associated outcomes are being proposed.[15–17] Associated outcomes are thought to include survival, quality of life, patient-reported outcomes, long-term effects, psychological and social outcomes.

BRIGHTLIGHT was a National Institute for Health Research funded programme of research with an overarching research question: 'Do Specialist Services for Teenagers and Young Adults with Cancer Add Value?'.[18 19] Specifically, it aimed to describe: what was age-appropriate care; what were the key components of such a service; what outcomes were impacted; and how much did it cost the National Health Service (NHS), young people and their families. Central to this programme was a cohort of young people aged 13–24 years at cancer diagnosis who were recruited within 4 months of a new cancer diagnosis and followed for 3 years: the BRIGHTLIGHT cohort.[18] We previously reported that survival of the BRIGHTLIGHT cohort was lower than the population diagnosed over the same time period but not recruited to the cohort, which was unrelated to cancer type.[18] We surmised that survival differences between the cohort and the non-recruited population may be related to the recruitment window of 4 months and therefore young people who were sicker had more contact with their treatment team and more opportunities for recruitment. The aim of this study was to determine whether there was any evidence of a causal association between the amount of care received in a TYA-PTC on survival outcomes and documentation of clinical processes of care.

## METHODS

### Study design

This paper reports results from several data sources: the longitudinal cohort study within BRIGHTLIGHT, a mixed methods programme of research, which obtained data from young people through a bespoke survey,[18] clinical report data completed by healthcare professionals and Demographic Batch Service data from NHS Digital.[18] The location of inpatient care for each participant was identified using routinely collected NHS Hospital Episodes Statistics (HES) data. We then developed a bespoke scale to assign each participant a category of care; see Taylor *et al* for detail.[18] Young people were assigned to a category dependent on the proportion of admitted patient care delivered in a TYA-PTC in the first 12 months after diagnosis: all care delivered in a TYA-PTC (ALL-TYA-PTC), no care in a TYA-PTC (NO-TYA-PTC) and some care delivered in a TYA-PTC with additional care in a children's or adult cancer unit (SOME-TYA-PTC). Sample size calculations were based on the primary outcome measure of the cohort, quality of life.[18]

### Participants and setting

The BRIGHTLIGHT cohort comprised young people aged 13–24 years, newly diagnosed with cancer (Classification of Disease ICD-10 codes C00-C97) in an English hospital and recruited within 4 months of diagnosis. Eligibility criteria were as inclusive as possible so no restriction according to language or any sensory impairment that affected communication was applied. The only exclusion criteria were: young people receiving a custodial sentence; if the young person was not anticipated to be alive at the first point of data collection (6 months after diagnosis); recurrence of a previous cancer or they were not capable of completing a survey, for example, sedated or in intensive care. The processes for recruitment are reported in detail elsewhere.[18 20 21] BRIGHTLIGHT was open to recruitment in 109 NHS hospitals in England, of whom 97 hospitals recruited at least one young person. Young people were recruited between October 2012 and April 2015 (diagnosed between July 2012 and December 2014). They gave written consent (parent consent also obtained from those less than 16 years); the study was

approved by London-Bloomsbury NHS Research Ethics Committee and the Confidentiality Advisory Group.

## Data collection

We assessed documentation of the following clinical processes:

► Histological diagnosis.
► Molecular confirmation of diagnosis (where relevant).
► Cancer stage or prognostic group defined (for leukaemia, total white blood cell count).
► Initial treatment plan.
► Treatment protocol for systemic therapy and/or for radiotherapy (RT).
► Evidence of MDT communication including children's, TYA or site-specific.
► Assessment by supportive care services based on evidence in notes of contact with a clinical nurse specialist plus one other member of the MDT (social worker, youth support coordinator, counsellor, psychologist, dietician, physiotherapist, occupational therapist).
► A record of fertility being discussed.
► A record of consideration for inclusion in a clinical trial.

Survival data were obtained from the Demographic Batch Service at NHS Digital reported up until October 2018.

To describe patient prognosis at recruitment, an existing scoring system was identified that uses anticipated 5-year survival to form groups of patients with expected survival of greater than 80%, 50%–80% and less than 50%.[22] To measure severity of illness, we developed a bespoke scale which accounts for the range of cancer types, staging systems, symptom burden, treatment burden, potential late effects and prognosis. This classifies patients as 'least severe', 'intermediate' and 'most severe' based on their cancer-specific information.[18]

## Analysis

Analysis was based on a predefined analysis plan using STATA V.15. For each clinical process outcome, the proportion of patients where the item was found to be documented in clinical records was reported by category of care. Proportions were compared across groups using $\chi^2$ tests (including a trend test).

Survival time for each participant was calculated from date of diagnosis to date of death or censored at the date last known to be alive up to 29 October 2018. Kaplan-Meier survival curves were plotted for each category of care (NO-TYA-PTC, SOME-TYA-PTC and ALL-TYA-PTC) and estimates of cumulative survival at 1–4 years (with 95% CIs) were calculated. The relationship between survival time and TYA category was investigated using a Cox regression model adjusted for confounding factors identified using a causal inference approach and based on the conceptual model underpinning the BRIGHT-LIGHT Survey[19] in the form of a Directed Acyclic Graph (DAG) (online supplemental file; DAGitty software www.

dagitty.net). Factors adjusted for were age at diagnosis, sex, type of cancer (leukaemia, lymphoma, brain and central nervous system (CNS), bone tumours, sarcoma, germ cell, melanoma, carcinomas, other), socioeconomic status (Index of Multiple Deprivation rank[23]), severity of cancer (least, intermediate, most[18]), ethnicity (white, other), days in hospital over 12 months since diagnosis, treatment received in 12 months since diagnosis (systemic anticancer therapy (SACT) only, RT only, Surgery only, Surgery and SACT/SACT and RT and Surgery/RT and SACT/Surgery and RT/Transplant, Other). Geographical region of treatment (North East, North West, Yorkshire, East Midlands, West Midlands, London, South East and South West) was included in the model as a random effect (frailty term). The proportional hazards assumptions of the Cox regression model were checked. Models were extended to include interaction terms to investigate whether the association between TYA group and survival was different by age at diagnosis (using categories of 13–18 and 19–24 years, and age in years) and tumour type (using categories, haematology and oncology).

## RESULTS

A total of 5953 incident cases were recorded in England, of which 5835 were eligible to participate and 1126 young people were recruited to the cohort (19.3%). Valid consent was available for 1114. Participation at each wave of data collection has been previously described.[18] Participant characteristics are shown in table 1. In comparison to TYA diagnosed in the same period but not recruited to the cohort, there was under-representation of patients with carcinoma, CNS cancers and melanoma, and over-representation of patients with leukaemia, lymphoma, germ cell tumours and bone tumours.[18] Two diagnostic groups accounted for 50% of the cohort, (lymphoma 31% and germ cell 19%).

Overall, 359 (34.3%) patients were in the NO-TYA-PTC group, 415 (39.8%) in the SOME-TYA-PTC group and 270 (25.9%) in the ALL-TYA-PTC. Cancer type varied by category of care, lymphoma was the most common in all groups (38%, 24%, 36%, NO, SOME, ALL, respectively). Leukaemia (20%) was the second most common cancer in the ALL-TYA-PTC, bone (22%) in the SOME-TYA-PTC and germ cell (20%) in the NO-TYA-PTC. There was variability in the distribution of prognosis and severity of illness scores between categories of care, the NO-TYA-PTC having the highest proportion of 'least severe disease' 70%, compared with 43% in the SOME-TYA-PTC and 49% in the ALL-TYA-PTC. The SOME-TYA-PTC had highest proportion of most severe disease, 33% compared with 11% and 22% in the NO and ALL groups, respectively. The NO-TYA-PTC group was also older (table 1).

The number of days in hospital over the 12 months since diagnosis varied between groups. For the NO-TYA-PTC group, the total number of days ranged from 1 to 213 (median 13, IQR, 4–27), for the SOME-TYA-PTC care group, total days ranged from 2 to 228 (median 59,

**Table 1** BRIGHTLIGHT cohort characteristics by level of teenagers and young adults (TYA) care category at 12 months from diagnosis

| Characteristic | Level of TYA care at 12 months from diagnosis | | |
| --- | --- | --- | --- |
| | NO-TYA-PTC N=359 | SOME-TYA-PTC N=415 | ALL-TYA-PTC N=270 |
| Age at diagnosis (years) | | | |
| Mean (SD) | 21.11 (3.04) | 19.44 (3.36) | 19.74 (3.23) |
| Gender | | | |
| Male | 193 (54%) | 224 (54%) | 156 (58%) |
| Female | 166 (46%) | 191 (46%) | 114 (42%) |
| Ethnicity* | N=351 | N=408 | N=259 |
| White | 312 (89%) | 344 (84%) | 221 (85%) |
| Mixed | 9 (3%) | 9 (2%) | 6 (2%) |
| Asian | 18 (5%) | 36 (9%) | 25 (10%) |
| Black | 7 (2%) | 11 (3%) | 2 (1%) |
| Chinese | 0 | 1 (<1%) | 2 (1%) |
| Other | 5 (1%) | 7 (2%) | 3 (1%) |
| Socioeconomic status (IMD quintile) | N=354 | N=404 | N=263 |
| 1—most deprived | 85 (24%) | 100 (25%) | 51 (20%) |
| 2 | 67 (19%) | 68 (17%) | 48 (18%) |
| 3 | 66 (19%) | 83 (21%) | 51 (19%) |
| 4 | 83 (23%) | 77 (19%) | 49 (19%) |
| 5—least deprived | 53 (15%) | 76 (19%) | 64 (24%) |
| Marital status | N=250 | N=262 | N=172 |
| Married/civil partnership | 9 (4%) | 8 (3%) | 6 (3%) |
| Cohabiting | 43 (17%) | 27 (10%) | 18 (10%) |
| Single/divorced | 198 (80%) | 227 (87%) | 148 (86%) |
| Current status | N=277 | N=312 | N=193 |
| Working full/part time | 126 (45%) | 72 (23%) | 43 (22%) |
| In education | 61 (22%) | 112 (36%) | 81 (42%) |
| Other work (apprentice/intern/voluntary) | 6 (2%) | 5 (2%) | 6 (3%) |
| Unemployed | 10 (4%) | 11 (4%) | 7 (4%) |
| Long-term sick | 39 (14%) | 51 (16%) | 31 (16%) |
| Not seeking work | 35 (13%) | 61 (19%) | 25 (13%) |
| Type of cancer[1] | | | |
| Leukaemia | 27 (8%) | 59 (14%) | 53 (20%) |
| Lymphoma | 138 (38%) | 100 (24%) | 96 (36%) |
| CNS | 12 (3%) | 13 (3%) | 17 (6%) |
| Bone | 10 (3%) | 93 (22%) | 9 (3%) |
| Sarcomas | 10 (3%) | 30 (7%) | 14 (5%) |
| Germ cell | 71 (20%) | 75 (18%) | 46 (17%) |
| Skin | 34 (9%) | 1 (<1%) | 6 (2%) |
| Carcinomas (not skin) | 51 (14%) | 41 (10%) | 27 (10%) |
| Miscellaneous specified† | 5 (1%) | 3 (<1%) | 1 (<1%) |
| Unspecified malignant | 1 (<1%) | 0 | 1 (<1%) |
| Severity of illness[18] | | | |
| Least | 251 (70%) | 180 (43%) | 131 (49%) |

Continued

**Table 1** Continued

| Characteristic | Level of TYA care at 12 months from diagnosis | | |
|---|---|---|---|
| | NO-TYA-PTC N=359 | SOME-TYA-PTC N=415 | ALL-TYA-PTC N=270 |
| Intermediate | 67 (19%) | 99 (24%) | 80 (30%) |
| Most | 41 (11%) | 136 (33%) | 59 (22%) |
| Prognostic score[23] | N=354 | N=413 | N=270 |
| <50% | 30 (8%) | 76 (18%) | 61 (23%) |
| 50%–80% | 70 (20%) | 166 (40%) | 65 (24%) |
| >80% | 254 (72%) | 171 (41%) | 144 (53%) |
| City‡ | N=359 | N=415 | N=270 |
| Birmingham | 54 (15%) | 75 (18%) | 18 (7%) |
| Bristol | 65 (18%) | 39 (9%) | 8 (3%) |
| Cambridge | 13 (4%) | 8 (2%) | 2 (1%) |
| Manchester | 32 (9%) | 44 (11%) | 20 (7%) |
| Merseyside | 15 (4%) | 13 (3%) | 11 (4%) |
| East Midlands | 19 (5%) | 34 (8%) | 73 (27%) |
| Leeds | 24 (7%) | 38 (9%) | 39 (14%) |
| Newcastle | 15 (4%) | 9 (2%) | 33 (12%) |
| Oxford | 6 (2%) | 5 (1%) | 8 (3%) |
| London | 84 (23%) | 116 (28%) | 14 (5%) |
| Sheffield | 8 (2%) | 13 (3%) | 13 (5%) |
| Southampton | 24 (7%) | 21 (5%) | 31 (11%) |
| Region‡ | N=359 | N=415 | N=270 |
| North East | 15 (4%) | 9 (2%) | 33 (12%) |
| North West | 47 (13%) | 57 (14%) | 31 (11%) |
| Yorkshire | 32 (9%) | 51 (12%) | 52 (19%) |
| East Midlands | 19 (5%) | 34 (8%) | 73 (27%) |
| West Midlands | 54 (15%) | 75 (18%) | 18 (7%) |
| London | 84 (23%) | 116 (28%) | 14 (5%) |
| South East | 43 (12%) | 34 (8%) | 41 (15%) |
| South West | 65 (18%) | 39 (9%) | 8 (3%) |
| Treatment received in the first 12 months since diagnosis | | | |
| SACT only | 111 (31%) | 114 (27%) | 119 (44%) |
| Surgery and SACT | 55 (15%) | 132 (32%) | 49 (18%) |
| Surgery only | 92 (26%) | 20 (5%) | 23 (9%) |
| SACT and RT | 49 (14%) | 61 (15%) | 30 (11%) |
| Surgery, RT and SACT | 12 (3%) | 60 (15%) | 24 (9%) |
| Surgery and RT | 17 (5%) | 9 (2%) | 16 (6%) |
| Transplant | 9 (3%) | 12 (3%) | 7 (3%) |
| RT only | 7 (2%) | 5 (1%) | 1 (<1%) |
| Other | 7 (2%) | 2 (<1%) | 1 (<1%) |
| Total days in hospital over 12 months | | | |
| Median (IQR), (max, min) | 13 (4–27) (1, 213) | 59 (20–103) (2, 228) | 29 (11–73) (1, 286) |
| Given a choice about where to receive treatment?§ | N=288 | N=356 | N=233 |

Continued

**Table 1**  Continued

| Characteristic | Level of TYA care at 12 months from diagnosis | | |
| --- | --- | --- | --- |
| | NO-TYA-PTC<br>N=359 | SOME-TYA-PTC<br>N=415 | ALL-TYA-PTC<br>N=270 |
| Yes | 121 (42%) | 86 (24%) | 48 (21%) |
| No (or <19 years) | 167 (58%) | 270 (76%) | 185 (79%) |
| Long-term condition prior to cancer? | N=277 | N=311 | N=193 |
| Yes | 20 (7%) | 34 (11%) | 18 (9%) |
| No | 257 (93%) | 277 (89%) | 175 (91%) |
| Time to diagnosis: days from first symptom | N=264 | N=304 | N=188 |
| Median (IQR), (min, max) | 62 (29.5–169.5)<br>(0, 1340) | 65.5 (29.5–152.5)<br>(0, 959) | 63.5 (25.5–151.0)<br>(0, 1217) |
| Time to diagnosis: number of GP visits before diagnosis | N=274 | N=311 | N=193 |
| Median (IQR), (min, max) | 1 (0–3)<br>(0, 20) | 1 (0–3)<br>(0, 20) | 2 (1–3)<br>(0, 40) |

Values are frequency (%) unless stated otherwise.
*Wave 1 data were used with missing values completed using available Public Health England data.
†Includes 4 'unclassified'—treated in cancer unit but did not have cancer.
‡Where available based on hospital of diagnosis, for 77 cases based on recruiting hospital. Note: Manchester=Christie,
Merseyside=Clatterbridge, London=the Royal Marsden Hospital/University College London Hospitals.
§Those <19 at diagnosis were assumed not to have been given a choice.
CNS, central nervous system; GP, general practitioner; IMD, Index of Multiple Deprivation; PTC, Principal Treatment Centres; RT, radiotherapy;
SACT, systemic anticancer therapy.

IQR 20–103) and for the ALL-TYA-PTC group the total number of days ranged from 1 to 286 (median 29, IQR 11–73).

## Processes of care

Clinical records were available for 1078 young people of which 1009 were assigned to: NO-TYA-PTC (n=333); SOME-TYA-PTC (n=409) and ALL-TYA-PTC (n=267). HES data were not available for 69 young people so they could not be assigned a category. The comparison of processes of care according to category of care is shown in table 2. There was no evidence of a difference between the three groups for the documentation of: histological diagnosis, cancer stage or prognosis, consideration for entry into a clinical trial and discussion at an MDT. Those receiving NO-TYA-PTC were more likely to have documented discussion in a site-specific MDT but had the lowest proportion with documented discussion in a TYA MDT and children's MDT. There was no significant difference between documentation of an initial treatment plan but there was a trend that this was more likely to have been recorded with more TYA-PTC care. Young people in NO-TYA-PTC had less frequent documentation of a molecular diagnosis (where molecular analysis was appropriate), discussions about fertility and assessments by supportive care services defined as contact with a clinical nurse specialist and one other professional such as youth support coordinator, social worker, psychologist (see methods for complete list).

## Survival

The duration of follow-up by October 2018 is shown in table 3. The number of deaths in the NO-TYA-PTC group was 27 (8%), compared with 35 (13%) in ALL-TYA-PTC and 91 (22%) in SOME-TYA-PTC. The cumulative probability of survival by time since diagnosis for the TYA-PTC categories is shown in figure 1 and table 4. Although survival probabilities at 1 year were similar, there was clear divergence between the groups over the following time period, such that probabilities were highest for those receiving NO-TYA-PTC, followed by ALL-TYA-PTC care, then SOME-TYA-PTC care. Following full adjustment for confounding factors, regression (table 5) showed there was no evidence of a relationship between the category of care and hazard (risk) of death.

Subgroup analyses showed no statistical evidence that the relationship between survival and level of care was different for the combined group of leukaemia and lymphomas compared with other cancers (table 6). There was however weak evidence of a difference in the effect of level of care on survival by age group, notably with lower risk of death when comparing SOME-TYA-PTC and ALL-TYA-PTC with NO-TYA-PTC in those aged under 19 years at diagnosis, while these relative risks were higher in the over 19 group. A similar pattern was seen when considering age as continuous.

**Table 2** Clinical process outcomes

| Documentation of: | NO-TYA-PTC N=333 | | | SOME-TYA-PTC N=409 | | | ALL-TYA-PTC N=267 | | | P value: $\chi^2$ trend |
|---|---|---|---|---|---|---|---|---|---|---|
| | N | Yes | No | N | Yes | No | N | Yes | No | |
| Histological diagnosis | 331 | 307 (93%) | 24 (7%) | 407 | 360 (88%) | 47 (12%) | 265 | 240 (91%) | 25 (9%) | 0.14 0.31 |
| Molecular diagnosis (where relevant)* | 186 | 49 (26%) | 137 (74%) | 304 | 106 (35%) | 198 (65%) | 200 | 87 (44%) | 113 (56%) | 0.002 0.02 |
| Cancer stage or prognostic group† | 333 | 311 (93%) | 22 (7%) | 409 | 383 (94%) | 26 (6%) | 267 | 253 (95%) | 14 (5%) | 0.77 0.50 |
| Initial treatment plan | 330 | 291 (88%) | 39 (12%) | 408 | 370 (91%) | 38 (9%) | 265 | 247 (94%) | 18 (7%) | 0.11 0.04 |
| Any MDT communication | 331 | 321 (97%) | 10 (3%) | 408 | 392 (96%) | 16 (4%) | 267 | 259 (97%) | 8 (3%) | 0.73 0.97 |
| Children's MDT | 329 | 34 (10%) | 295 (90%) | 403 | 81 (20%) | 322 (80%) | 265 | 58 (22%) | 207 (78%) | <0.001 <0.001 |
| TYA MDT | 326 | 164 (50%) | 162 (50% | 401 | 285 (71%) | 116 (29%) | 265 | 207 (78%) | 58 (22%) | <0.001 <0.001 |
| Site-specific MDT | 325 | 271 (83%) | 54 (17%) | 402 | 285 (71%) | 117 (29%) | 264 | 189 (72%) | 75 (28%) | <0.001 0.001 |
| Assessment by supportive care services | 327 | 124 (38%) | 203 (62%) | 405 | 249 (61%) | 156 (39%) | 258 | 154 (60%) | 104 (40%) | <0.001 <0.001 |
| Fertility being discussed (all) | 330 | 178 (54%) | 152 (46%) | 407 | 282 (69%) | 125 (31%) | 259 | 195 (75%) | 64 (25%) | <0.001 <0.001 |
| Fertility discussed (males) | 178 | 112 (63%) | 66 (27%) | 221 | 172 (78%) | 49 (22%) | 152 | 117 (77%) | 35 (23%) | 0.002 0.003 |
| Fertility discussed (females) | 152 | 66 (43%) | 110 (59%) | 186 | 110 (59%) | 76 (41%) | 107 | 78 (73%) | 29 (27%) | <0.001 <0.001 |
| Consideration into clinical trial | 328 | 207 (63%) | 121 (37%) | 405 | 252 (62%) | 153 (38%) | 256 | 176 (69%) | 80 (31%) | 0.21 0.19 |

From case report form data: completed/partially completed=1078; 1009 have category of specialist care recorded.
*indicated as 'not relevant' for: NO-teenage and young adult (TYA)-Principal Treatment Centres (PTC), n=137; SOME-TYA-PTC, n=97; ALL-TYA-PTC, n=65.
†Cancer stage or prognostic group documented is defined as: for leukaemia—a white blood cell count measure is provided; for lymphoma if stage (1–4) is entered (variable 'stage'); for solid tumour use variable 'has the tumour been staged?' If these things are not recorded for the appropriate cancer type, then coded as not documented. Cancer type is determined by birch classification.
MDT, multidisciplinary team.

## DISCUSSION

We have reported on a national longitudinal evaluation of specialist cancer services for young people aged 13–24 years at diagnosis defining the TYA-PTCs and their networks as they were described in the UK NICE Improving Outcomes Guidance in 2005.[12] We used routinely collected NHS data (HES) which records hospital admission data to measure how much care young people received in the TYA-PTC, dividing our cohort into three distinct groups, all care delivered in a TYA-PTC (ALL-TYA-PTC) no care in a TYA-PTC (NO-TYA-PTC) and those who received some care in the TYA-PTC and other parts of their care in another children's or adult hospital. We assessed documentation of clinical processes

**Table 3** Duration of follow-up

| | NO-TYA-PTC N=359 | SOME-TYA-PTC N=415 | ALL-TYA-PTC N=270 | TOTAL N=114 |
|---|---|---|---|---|
| Median (IQR) follow-up (days) | 1839 (1597–2041) | 1743 (1474–1991) | 1747 (1536–2023) | 1779 (1536–2023) |

PTC, Principal Treatment Centres; TYA, teenagers and young adults.

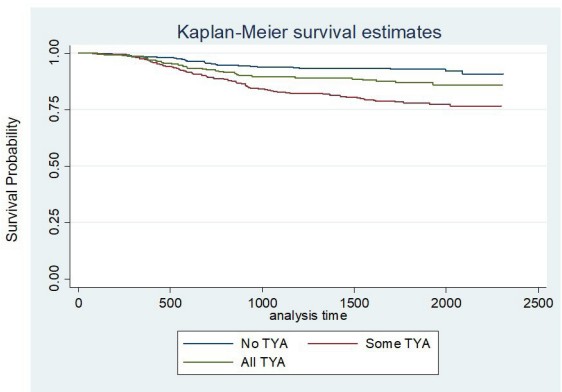

**Figure 1** Comparison of survival according to the three categories of care (unadjusted). TYA, teenagers and young adults.

assumed to be related to quality of care and found those receiving NO-TYA-PTC were less likely to have a record of molecular diagnosis (where relevant). Additionally, this group were less likely to have documentation of review by a children's or TYA MDT, have an assessment by supportive care services or have a fertility discussion compared with those treated in SOME-TYA-PTC or ALL-TYA-PTC. These are criteria which we would expect to be associated with specialist age-appropriate care[16 17]; therefore, it is not surprising these appear to be more frequently documented in the ALL and SOME group.

Our results suggest differences between the groups in these measures of the quality of cancer care delivered to young people. However, this did not appear to have an impact on survival outcomes. One-year survival was similar between the groups; at 4 years, survival was highest in the NO-TYA-PTC group, followed by ALL-TYA-PTC and lowest in the SOME-TYA-PTC group. However, this was not significant. One of the reasons for implementing a new model of care in 2005 specific for TYA with cancer was due to the disparity in survival compared with children and older adults.[12] It is therefore disappointing that there were no survival differences noted between the three categories. Alternatively, we could view this as a positive finding—wherever young people choose to be treated, their survival outcomes are the same. The work we did with young people to develop BRIGHTLIGHT highlighted that they did not perceive survival alone as the most import outcome. Quality of life and the ability to get back on with life were as important.[18 19] While we

have shown a better improvement in quality of life when treated in a TYA-PTC,[23] we have yet to ascertain whether young people's reintegration into life when treatment ends is also better.

There was weak evidence that increasing age was associated with higher risk of death for those in the SOME-TYA-PTC compared with NO-TYA-PTC and ALL-TYA-PTC, an important finding given that it is this group who have a choice over where to receive their care.[12] We believe that further investigation into the lower survival in the SOME-TYA-PTC group is warranted, particularly the association with age. Those aged over 19 years at diagnosis had the more pronounced effects in the subgroup analysis, but the direction of the effect differed substantially for under 19s compared with the overall effect.

There is a paucity of existing literature to compare our results with and comparisons are further confounded by variation in healthcare systems, distinct models of specialist age-appropriate care adopted and the international definition of TYA which can extend up to 39 years in some countries.[24] A previous retrospective regional study of children and TYA in England found a survival benefit of being treated in a PTC for poor prognosis leukaemia and a converse relationship for those with soft tissue sarcoma, no significant differences in survival were observed for those with lymphoma, CNS, bone and germ cell tumours.[25] Of note, a previous study has also shown those receiving 'SOME' specialist care have poorer survival for some indications (Birch 2013, unpublished thesis). These studies suggest that some tumour groups may benefit from care at the PTC however, due to our previously reported difficulties with recruitment and reduction in sample size[20] we were unable to conduct the detailed analysis of individual cancer types as planned, and thus benefits of the PTC may be masked within the grouping of 'haematology' and 'solid tumours'.

## Limitations
Despite our study including a large, broadly representative sample of newly diagnosed TYA with cancer followed-up for 3 years, and our analyses being adjusted for factors known to affect outcomes in cancer there are some limitations to our study. Our definition of 'specialist' was based on the TYA-PTC care model as defined by the NICE Improving Outcomes Guidance issued in 2005,[12] which does not necessarily reflect current delivery

**Table 4** Estimated cumulative survival probabilities by categories of teenagers and young adults (TYA) care and year from diagnosis (95% CI)

|  | NO-TYA-PTC | SOME-TYA-PTC | ALL-TYA-PTC |
|---|---|---|---|
| 1 year | 0.98 (0.96 to 0.99) | 0.97 (0.95 to 0.99) | 0.98 (0.95 to 0.99) |
| 2 years | 0.95 (0.92 to 0.97) | 0.89 (0.86 to 0.92) | 0.93 (0.89 to 0.95) |
| 3 years | 0.94 (0.91 to 0.96) | 0.83 (0.79 to 0.86) | 0.90 (0.85 to 0.93) |
| 4 years | 0.93 (0.90 to 0.95) | 0.80 (0.76 to 0.84) | 0.89 (0.84 to 0.92) |

PTC, Principal Treatment Centres.

**Table 5**  Results from Cox regression model for survival from diagnosis by categories of teenagers and young adults (TYA) care received during the first 12 months from diagnosis

| | | HR | 95% CI | P value * |
|---|---|---|---|---|
| Unadjusted model (N=1044) | | | | |
| TYA care category (vs NO-TYA-PTC) | SOME-TYA-PTC | 3.14 | 2.04 to 4.83 | p<0.001 |
| | ALL-TYA-PTC | 1.79 | 1.08 to 2.96 | |
| Fully adjusted model (N=1000) | | | | |
| TYA care category (vs NO-TYA-PTC) | SOME-TYA-PTC | 1.55 | 0.94 to 2.58 | p=0.15 |
| | ALL-TYA-PTC | 1.13 | 0.64 to 1.97 | |

*P value from a likelihood ratio test.
†Adjusted for age at diagnosis, sex, type of cancer, socioeconomic status (Index of Multiple Deprivation rank), severity of cancer, treatment, days in hospital, and ethnicity geographical region of treatment were included as a random effect (frailty term).
PTC, Principal Treatment Centres.

of age-appropriate care.[9] The study population were recruited during a period of evolution of TYA services in England; therefore, the models of care are unlikely to reflect current practice, particularly as we have identified that specialist age-appropriate care takes time to develop (Lea 2019 Unpublished thesis).[13] Additionally, categorising TYA-PTC assumes that all PTCs are equal and does not measure the quality of care delivered. We know national variation exists in configuration and maturity of services,[26] particularly during 2012–2014 when patients were recruited. Further, due to the coding of hospital inpatient data it is possible that some patients have been misattributed as receiving care in the TYA-PTC when they

may have been cared for in a Trust which had a TYA-PTC but care was delivered at a different hospital and not in the TYA unit. An additional limitation of the categorisation of care was that it was based on previous work (Birch 2013, unpublished thesis), which only included inpatient admission data. Potentially, hospital visits involving treatment as outpatient care were not included. This could have resulted in patients been misclassified as ALL-TYA-PTC or NO-TYA-PTC.

Consideration must also be given to additional factors influencing survival outcomes which we did not collect or measure. These include deviation from the intended treatment plan such as the proportion of treatment

**Table 6**  Planned subgroup investigations for cancer type (leukaemia/lymphoma vs other) and age group (<19 vs 19+): results from fully adjusted* models with interaction terms (N=1000)

| | TYA care category | Fully adjusted HR | 95% CI | P value from interaction |
|---|---|---|---|---|
| Cancer type | | | | |
| Leukaemia/lymphoma | SOME-TYA-PTC versus NO-TYA-PTC | 1.37 | 0.63 to 3.01 | p=0.95 |
| | ALL-TYA-PTC versus NO-TYA-PTC | 0.97 | 0.41 to 2.28 | |
| Other cancers | SOME-TYA-PTC versus NO-TYA-PTC | 1.34 | 0.68 to 2.63 | |
| | ALL-TYA-PTC versus NO-TYA-PTC | 1.09 | 0.52 to 2.27 | |
| Age group | | | | |
| Age<19 years | SOME-TYA-PTC versus NO-TYA-PTC | 0.81 | 0.41 to 1.57 | p=0.08 |
| | ALL-TYA-PTC versus NO-TYA-PTC | 0.79 | 0.37 to 1.71 | |
| Age 19+years | SOME-TYA-PTC versus NO-TYA-PTC | 1.75 | 0.99 to 3.06 | |
| | ALL-TYA-PTC versus NO-TYA-PTC | 1.14 | 0.59 to 2.23 | |
| Continuous age | | Coefficient for age (per year) | | |
| | NO-TYA-PTC | 0.95 | 0.85 to 1.06 | p=0.07 |
| | SOME-TYA-PTC | 1.11 | 1.03 to 1.18 | |
| | ALL-TYA-PTC | 1.05 | 0.94 to 1.17 | |

Adjusted for age at diagnosis, type of cancer (detailed categories), socioeconomic status (Index of Multiple Deprivation rank), severity of cancer, ethnicity (white vs other), gender, treatment (detailed categories) received in 6 months from diagnosis, days in hospital within 12 months of diagnosis with region as random effect.
PTC, Principal Treatment Centres; TYA, teenagers and young adults.

delivered, delays/reduction in delivery and toxicity. Therefore, it was not possible to determine the dose or type of chemotherapy or RT received by patients in each group and these would be important determinants of survival. Our scale for place of care was derived only from inpatient care and not care or treatment delivered as an outpatient. Thus, we may have missed considerable elements of the care received.

Further to this, overall survival of the BRIGHTLIGHT cohort was lower than those diagnosed during the same period but not recruited; therefore, our findings may not reflect the experience of the whole TYA population.[18] We also do not know the decision-making processes behind referral of patients into each TYA-PTC group at diagnosis, it may be that patients with better prognosis are treated more locally with site specific expertise competent at treating the cancer with good survival outcomes, while those with more complex disease and holistic needs are referred into the specialist TYA service.

## CONCLUSION

We have reported on the first systematic longitudinal evaluation of cancer services for young people. Young people were more likely to have had documentation of access to supportive care services or have a fertility discussion if they had some or all of their care delivered by the TYA-PTC, which existing literature supports as important for TYA.

Overall, survival at 4 years was good across all three categories of care with some differences between the NONE, ALL and SOME groups as defined by NICE improving outcomes service specification in 2005. The factors contributing to survival differences between the groups warrants further investigation particularly the relationship between survival, level of TYA care and age. BRIGHTLIGHT results are immediately important for current healthcare recommendations for young people with cancer in England. The currently proposed model of care proposed by NHS England advocates 'Joint Care' but with an emphasis to increase communication between the TYA-PTCs and selected local hospitals. Further enquiry is required with additional prospective data collection to assess whether this proposed Joint Care would generate a similar pattern of survival trends as the 'SOME-TYA-PTC' group in our study.

**Author affiliations**
[1]Oncology, University College London Hospitals NHS Foundation Trust, London, UK
[2]Centre for Nurse, Midwife and AHP Led Research (CNMAR), University College London Hospitals NHS Foundation Trust, London, UK
[3]Department of Statistical Science, University College London, London, UK
[4]Department of Biomedicine, Biotechnology and Public Health, University of Cadiz, Cadiz, Spain
[5]School of Medicine, University of Leeds, Leeds, UK
[6]Cancer Service, University College London Hospitals NHS Foundation Trust, London, UK
[7]Cancer Clinical Trials, University College London Hospitals NHS Foundation Trust, London, UK
[8]Primary Care Unit, University of Cambridge, Cambridge, UK
[9]Wessex Teenage and Young Adult Cancer Service, University Hospital Southampton NHS Foundation Trust, Southampton, UK
[10]Faculty of Health and Medical Sciences, University of Surrey, Guildford, UK
[11]Centre for Outcomes and Experience Research in Children's Health, Illness and Disability (ORCHID), Great Ormond Street Hospital For Children NHS Foundation Trust, London, UK
[12]Institute of Epidemiology & Health, University College London, London, UK
[13]Leeds Insitute of Molecular Medicine, University of Leeds, Leeds, UK

**Acknowledgements** We would like to thank the members of our Young Advisory Panel (Zeena Beale, Ciaran Fenton, Emily Freemantle, Jaasjan Guvindia, Laura Haddard, Steph Hammersley, Amy Lang, Joshua Lerner, Tanya Loughlin, Jason, Sin Jin Loo, Jennifer Miller, Maria Onasanya, Arif Nasir, Steph Still, Poppy Richards, Amy Riley, Lara Veitch, Freya Voss, JJ Wheeler, Max Willliamson, Antonia Young), the 1,114 young people who consented to participate in BRIGHTLIGHT, healthcare professionals who approached and consented young people, and ex-members of the team who have contributed to study management (Catherine O'Hara, Anita Solanki, Natasha Aslam, Zuwena Fox). We would like to dedicate this manuscript in memory of Mr Stephen Sutton and Mr Mathew Cook who were instrumental to study set up, design and management. Both of whom died from their cancer during the study. We would also like to thanks the following for all their support with recruitment to BRIGHTLIGHT: the National Cancer Research Institute, especially Dr Eileen Loucaides and the Secretariat; Matt Seymour, Matt Cooper and Karen Poole at the former National Cancer Research Network; Maria Khan and Sabrina Sandhu (North West Knowledge Intelligence Team); TYAC; Teenage Cancer Trust; CLIC Sargent; Ipsos MORI; Quality Health and the research teams at 109 NHS Trusts in England who opened BRIGHTLIGHT to recruitment.

**Collaborators** Principal Investigators agreeing to be acknowledged for their contribution to BRIGHTLIGHT recruitment: Claire Hemmaway, Barking, Havering and Redbridge Hospitals NHS Trust; Anita Amadi, Barnet and Chase Farm Hospitals NHS Trust; Keith Elliott, Barnsley Hospital NHS Foundation Trust; Leanne Smith, Blackpool, Fylde and Wyre Hospitals NHS Trust; Shirley Cocks, Bolton NHS Foundation Trust; Victoria Drew, Bradford Teaching Hospitals NHS Foundation Trust; Elizabeth Pask, Central Manchester University Hospitals NHS Foundation Trust; Anne Littley, Central Manchester University Hospitals NHS Foundation Trust; Mark Bower, Chelsea and Westminster Hospital NHS Trust; Scott Marshall, City Hospitals Sunderland NHS Foundation Trust; Lorna Dewar, Colchester Hospital University NHS Trust; Nnenna Osuji, Croydon Health Services NHS Trust; David Allotey, Derby Hospitals NHS Foundation Trust; Karen Jewers, East Lancashire Hospitals NHS Trust; Asha Johny, Gloucestershire Hospitals NHS Foundation Trust; Nicola Knightly, Great Western Hospitals NHS Foundation Trust; Robert Carr, Guy's & St Thomas' Hospital NHS Foundation Trust; Alison Milne, Hampshire Hospitals NHS Foundation Trust; Claire Hall, Harrogate and District NHS Foundation Trust; James Bailey, Hull and East Yorkshire Hospitals NHS Trust; Christine Garlick, Ipswich Hospital NHS Foundation Trust; Alison Brown, Isle of Wight Healthcare NHS Trust; Carolyn Hatch, Lancashire Teaching Hospitals NHS Foundation Trust; Vivienne E. Andrews, Medway NHS Foundation Trust; Sara Greig, Milton Keynes Hospital NHS Foundation Trust; Jennifer Wimperis, Norfolk and Norwich University Hospital NHS Trust; Suriya Kirkpatrick, North Bristol NHS Trust; Jonathan Nicoll, North Cumbria University Hospitals NHS Trust; Ivo Hennig, Nottingham University Hospitals NHS Trust; Karen Sherbourne, Oxford Radcliffe Hospital NHS Trust; Clare Turner, Plymouth Hospitals NHS Trust; Claire Palles-Clark, Royal Surrey County Hospital NHS Trust; Christine Cox, Royal United Hospital Bath NHS Trust; Yeng Ang, Salford Royal NHS Foundation Trust; Jonathan Cullis, Salisbury NHS Foundation Trust; Daniel Yeomanson, Sheffield Children's NHS Foundation Trust; Ruth Logan, Sheffield Teaching Hospitals NHS Foundation Trust; Deborah Turner, South Devon Healthcare NHS Trust; Dianne Plews, South Tees Hospitals NHS Trust; Juliah Jonasi, Southend University Hospital NHS Foundation Trust; Ruth Pettengell, St George's Healthcare NHS Trust; Kamal Khoobarry, Surrey & Sussex Healthcare NHS Trust; Angela Watts, The Dudley Group of Hospitals NHS Foundation Trust; Louise Soanes, The Royal Marsden NHS Foundation Trust; Claudette Jones, The Royal Orthopaedic Hospital NHS Trust; Michael Jenkinson, The Walton Centre for Neurology and Neurosurgery NHS Trust; Nicky Pettitt, University Hospital Birmingham NHS Foundation Trust; Vijay Agarwal, University Hospital Birmingham NHS Foundation Trust; Beth Harrison, University Hospitals Coventry and Warwickshire NHS Trust; Fiona Miall, University Hospitals of Leicester NHS Trust; Gail Wiley, University Hospitals of Morecambe Bay NHS Trust; Lynda Wagstaff, Walsall Hospitals NHS Trust; Fiona Smith, West Hertfordshire Hospitals NHS Trust; Sarah Janes, Western Sussex NHS Trust; Serena Hillman, Weston Area Health NHS Trust; Christopher Zaborowski, Yeovil District Hospital NHS Foundation Trust. Data for this study is based on information collected and quality

assured by the PHE National Cancer Registration and Analysis Service. Access to the data was facilitated by the PHE Office for Data Release.

**Contributors** RMT, LAF, JB, DPS, SM, RF, LH, FG, RR, JW were involved in developing the protocol. RMT coordinated the running of the study and was responsible for data acquisition. JB, JA-G, RMT, LAF, AM, SL, SM, RF, DPS, JW contributed to the analysis. RMT, LAF, JB and JW drafted the manuscript. All authors critically revised and approved the final manuscript.

**Funding** This paper presents independent research funded by the National Institute for Health Research (NIHR) under its Programme Grants for Applied Research Programme (Grant Reference Number RP-PG-1209-10013). The views expressed are those of the author(s) and not necessarily those of the NIHR or the Department of Health and Social Care. The BRIGHTLIGHT Team acknowledges the support of the NIHR, through the Cancer Research Network. LAF and LH are funded by Teenage Cancer Trust, DS holds research grant funding from Teenage Cancer Trust, and RR was (in part) supported by the National Institute for Health Research (NIHR) Collaboration for Leadership in Applied Health Research and Care (CLAHRC) North Thames at Bart's Health NHS Trust. RMT is a National Institute for Health Research (NIHR) Senior Nurse Research Leader. The views expressed in this article are those of the author and not necessarily those of the NIHR, or the Department of Health and Social Care. JAG was subsidised by the Ramon & Cajal programme operated by the Ministry of Economy and Business (RYC-2016-19353), and the European Social Fund. None of the funding bodies have been involved with study concept, design or decision to submit the manuscript.

**Competing interests** None declared.

**Patient and public involvement statement** Young people have been involved in this study from the feasibility stage onward. They were involved in study development, acted as co-researchers and were instrumental in the design and methods of the study. A representative of the Young Advisory Panel (YAP) was a co-applicant on the grant and the YAP have been part of the management of the study since the grant was awarded in 2011. Details of the extent of young people's involvement in BRIGHTLIGHT is provided in reference 13.

**Patient consent for publication** Not required.

**Ethics approval** The study was approved by the Health Research Authority Confidentiality Advisory Group (reference ECC 8-05(d)/2011) and London Bloomsbury NHS Research Ethics Committee (reference LO/11/1718).

**Provenance and peer review** Not commissioned; externally peer reviewed.

**Data availability statement** Data are available upon reasonable request. Data that are not held under licence with Public Health England or NHS Digital will be available when the primary analysis is complete. We welcome collaboration, for general data sharing enquiries please contact RMT (rtaylor13@nhs.net).

**ORCID iDs**
Rachel M Taylor http://orcid.org/0000-0002-0853-0925
Javier Alvarez-Galvez http://orcid.org/0000-0001-9512-7853
Stephen Morris http://orcid.org/0000-0002-5828-3563
Jeremy Whelan http://orcid.org/0000-0001-6793-5722

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
