## [Reviewer comments · BMJ Open]

ARTICLE DETAILS

TITLE (PROVISIONAL)	Processes of care and survival associated with treatment in specialist teenage and young adult cancer centres: results from the BRIGHTLIGHT cohort study
AUTHORS	Fern, Lorna; Taylor, Rachel; Barber, Julie; Alvarez-Galvez, Javier; Feltbower, Richard; Lea, Sarah; Martins, Ana; Morris, Stephen; Hooker, Louise; Gibson, Faith; Raine, Rosalind; Stark, Dan; Whelan, Jeremy

VERSION 1 – REVIEW

REVIEWER	Michael Osborn Youth Cancer Service, Royal Adelaide Hospital AUSTRALIA
REVIEW RETURNED	09-Dec-2020

GENERAL COMMENTS	Thank you for the opportunity to review this manuscript resulting from the Brightlight study, focussing on whether exposure to TYA-PTC is associated with survival and documentation of clinical processes of care. I found the article to be interesting, balanced and well-written. It also addressed some of the questions that I was interested in after reading the authors' previous manuscript on the primary outcome of the Brightlight study. I cannot identify any major problems with this manuscript, but noted the following minor issues: 1. There appear to be a couple of typo's in the Methods section:- Page 6 Line 16: Should "...a casual inference approach..." be "...a causal inference approach..."?- Page 6 Line 30: There shouldn't be an apostrophe in "hazards" (ie. it should be "hazards" not "hazard's")2. In the discussion, the paragraph prior to the conclusion (Page 17 Lines 19-25) needs editing. At the very least, the commas in the second and fourth lines of this paragraph should be full stops (or appropriate conjunctions added). If I am being critical, you may wish to re-write this paragraph to make the concepts a little clearer for the casual reader.3. The final sentence of the conclusion has a comma which I think should be a fullstop (Page 17 line 38).4. I would be interested to hear more about the concept raised in this final sentence. As you mention, the currently proposed model advocates "Joint Care", but do you think that your data supports this? In your previous manuscript, the SOME group had the lowest quality of life (although I acknowledge that their quality of life did
--

	improve at a comparable rate to the ALL group). In this current manuscript, some of the processes of care appear to be better in the SOME group than the NO group, but the raw numbers are still not as good as the ALL group. Additionally, for the >19 year olds there was weak evidence of worse survival in the SOME group in the adjusted model. With all of that in mind, do you think that the data supports "joint care", or have other factors (?economic, ?political, ?reluctance of young people to travel for care) driven this policy decision? Should this be re-considered in light of your data, or have the models of care evolved so much that current joint care looks very different to that experienced by the SOME group in 2012-2014? There may be good reasons not to publically express your opinions on whether Joint Care is the best model for the future in this article (and I certainly don't think this has to be added to the current manuscript), but I am still interested to hear your view. (Please note that I don't mind if you don't address this point in your revised manuscript) In conclusion, this is a very good manuscript, and once the minor issues raised in dot-points 1-3 above have been addressed, I would support its publication.
--	--

REVIEWER	Sumit Gupta Hospital for Sick Children Canada
REVIEW RETURNED	09-Dec-2020

GENERAL COMMENTS	The authors present the results of the BRIGHTLIGHT study, which as one of its objectives attempted to determine the impact of their TYA specific centers. Their cohort was divided into those TYA who received all their care at a TYA specific centre, those who received some care there, and those that received no care there. Differences were seen in specific processes of care (e.g. more fertility referral documentation) but not in survival outcomes. This is a very important study; TYA care has received increasing attention in the last 2 weeks given this population's vulnerability. Many systems and providers have embraced the mantra that specialty care should be standard for AYA, but evidence supporting this is lacking. This contribution by Dr. Taylor et al is therefore very welcome. I do however have several concerns and questions that I believe would strengthen the manuscript if addressed: MAJOR COMMENTS  1. Inpatient vs. outpatient care – Patient treatment setting was categorized according to where admitted care was delivered. Does this mean that the location of outpatient care was not accounted for? If so, as many AYA cancers (Hodgkin, some NHL, some solid tumours, GCT) are treated on an outpatient basis, how were patients who were not admitted categorized? Second, for a specific patient, may outpatient and inpatient care have been delivered in different settings, through the use of say satellite or community centres? If these are the case, this may introduce misclassification into the patient's locus of care categorization. 2. Differences in characteristics by level of TYA care category are described in Table 1. It is difficult to determine which of these differences is statistically significant. A multivariable logistic
--

	regression will be difficult given that the outcome has three categories, but some idea of how these characteristics vary, or confound each other, may be useful. 3. Residual confounding by cancer type – Though the sample size is likely insufficient to categorize cancer type any more finely, it would be important to acknowledge as a limitation that residual confounding likely exists within the current classification. AML vs. ALL, or HL vs. NHL for example, may well have different findings from each other when looking at difference between treatment setting, that would be obscured. 4. Discussion – I feel the Discussion underplays the significance of the non survival outcomes being different by treatment care setting. There is a rich literature on the importance of psychosocial care and fertility discussions to AYA, so even if the finding of no survival differences is true, this alone may be a point that demonstrated the value of these TYA centres. This warrants much more discussion than it currently receives in the manuscript. 5. The finding of possible differences in survival among older BRIGHTLIGHT AYA is intriguing. The authors note that this group (>19 years) is theoretically given a choice as to where their treatment is received. Another factor which perhaps should be discussed is that the alternative for younger TYA to TYA-PTC will likely be pediatric cancer centres, while that for ≥ 19yo will be adult centres. Many studies have shown that even in systems without AYA specific units/centres, care differs between these two treatment settings for AYA. Is it possible to perform a sensitivity analysis where patients in the NONE or SOME categories are excluded if their non PTC care was received in a pediatric centre? I realize that this may represent additional work for the authors, but may be a very important finding. MINOR COMMENTS 1. Table 3 – Consider renaming the ALL category to avoid confusion between ALL and ALL-TYA-PTC.
--	--

VERSION 1 – AUTHOR RESPONSE

Reviewer: 1

Dr. Michael Osborn, Royal Adelaide Hospital

Comments to the Author:

Thank you for the opportunity to review this manuscript resulting from the Brightlight study, focussing on whether exposure to TYA-PTC is associated with survival and documentation of clinical processes of care. I found the article to be interesting, balanced and well-written. It also addressed some of the questions that I was interested in after reading the authors' previous manuscript on the primary outcome of the Brightlight study. I cannot identify any major problems with this manuscript, but noted the following minor issues:

Dear Michael, thank you for your kind comments, we are glad you found the manuscript interesting. We hope we have addressed your minor corrections cited below.

1. There appear to be a couple of typo's in the Methods section:

- Page 6 Line 16: Should "...a casual inference approach..." be "...a causal inference approach...?"
- Page 6 Line 30: There shouldn't be an apostrophe in "hazards" (ie. it should be "hazards" not "hazard's")

Thank you for pointing out these typo's, we have corrected accordingly.

2. In the discussion, the paragraph prior to the conclusion (Page 17 Lines 19-25) needs editing. At the very least, the commas in the second and fourth lines of this paragraph should be full stops (or appropriate conjunctions added). If I am being critical, you may wish to re-write this paragraph to make the concepts a little clearer for the casual reader.

Thank you for pointing this out, we had omitted to note the whole paragraph had very little punctuation. We have re-written it as you suggested.

3. The final sentence of the conclusion has a comma which I think should be a fullstop (Page 17 line 38).

We have amended accordingly.

4. I would be interested to hear more about the concept raised in this final sentence. As you mention, the currently proposed model advocates "Joint Care", but do you think that your data supports this? In your previous manuscript, the SOME group had the lowest quality of life (although I acknowledge that their quality of life did improve at a comparable rate to the ALL group). In this current manuscript, some of the processes of care appear to be better in the SOME group than the NO group, but the raw numbers are still not as good as the ALL group. Additionally, for the >19 year olds there was weak evidence of worse survival in the SOME group in the adjusted model. With all of that in mind, do you think that the data supports "joint care", or have other factors (?economic, ?political, ?reluctance of young people to travel for care) driven this policy decision? Should this be re-considered in light of your data, or have the models of care evolved so much that current joint care looks very different to that experienced by the SOME group in 2012-2014? There may be good reasons not to publically express your opinions on whether Joint Care is the best model for the future in this article (and I certainly don't think this has to be added to the current manuscript), but I am still interested to hear your view. (Please note that I don't mind if you don't address this point in your revised manuscript)

This is a very good comment as it raises an important point as to where the recommendation for 'joint care' comes from. We do not advocate joint care based on the data from our cohort study, the point we were trying to raise was that 'joint care' is advocated by the new proposed NHS service specification for teenagers and young adults with cancer in England. We have added some clarity around this statement in the concluding paragraph: 'The currently proposed model of care proposed by NHS England advocates 'Joint Care' but with an emphasis to increase communication between the TYA-PTCs and selected local hospitals. Further enquiry is required with additional prospective data collection to assess whether this proposed 'Joint Care' would generate a similar pattern of survival trends as the 'SOME-TYA-PTC' Group in our study'.

We are indeed incredibly concerned about this recommendation given the findings of the BRIGHTLIGHT cohort study. However, you raise a good point about the evolution of services and whether the data generated in 2012-2014 is relevant and generalisable with the current population of TYA accessing services. Our case study conducted in 2015 that formed part of the BRIGHTLIGHT programme grant very much highlighted how the culture of TYA services required time to develop and how services evolve over time. In this light, we have secured additional funding from the NIHR to

carry out a rapid evaluation of current services, albeit this is somewhat delayed now due to impact on COVID on NHS hospitals. We have not included this in the manuscript as we do not want to detract from the main messages.

In conclusion, this is a very good manuscript, and once the minor issues raised in dot-points 1-3 above have been addressed, I would support its publication.

Thank you for your comments, particularly around who was proposing 'joint care' as we would not make this recommendation from our dataset.

Reviewer: 2

Dr. Sumit Gupta, Hosp Sick Children

Comments to the Author:

The authors present the results of the BRIGHTLIGHT study, which as one of its objectives attempted to determine the impact of their TYA specific centers. Their cohort was divided into those TYA who received all their care at a TYA specific centre, those who received some care there, and those that received no care there. Differences were seen in specific processes of care (e.g. more fertility referral documentation) but not in survival outcomes. This is a very important study; TYA care has received increasing attention in the last 2 weeks given this population's vulnerability. Many systems and providers have embraced the mantra that specialty care should be standard for AYA, but evidence supporting this is lacking. This contribution by Dr. Taylor et al is therefore very welcome. I do however have several concerns and questions that I believe would strengthen the manuscript if addressed:

Thank you for the review of our manuscript, we hope the response to your comments alleviates some of your concerns.

MAJOR COMMENTS

1. Inpatient vs. outpatient care – Patient treatment setting was categorized according to where admitted care was delivered. Does this mean that the location of outpatient care was not accounted for? If so, as many AYA cancers (Hodgkin, some NHL, some solid tumours, GCT) are treated on an outpatient basis, how were patients who were not admitted categorized? Second, for a specific patient, may outpatient and inpatient care have been delivered in different settings, through the use of say satellite or community centres? If these are the case, this may introduce misclassification into the patient's locus of care categorization.

Thank you for your comment and you raise a very valid point about the limitations of the current scale used to measure the amount of care delivered in the TYA-PTC. It is possible that some patients may have been misclassified as we only used admitted patient care data as we were following a methodology previously used in this population. We have added this as a limitation : 'Further, due to the coding of hospital inpatient data it is possible that some patients have been misattributed as receiving care in the TYA-PTC when they may have been cared for in a Trust which had a TYA-PTC but care was delivered at a different hospital and not in the TYA unit. An additional limitation of the categorisation of care was that it was based on previous work (26), which only included inpatient admission data. Potentially, hospital visits involving treatment as outpatient care were not included. This could have resulted in patients been misclassified as ALL-TYA-PTC or NO-TYA-PTC.

2. Differences in characteristics by level of TYA care category are described in Table 1. It is difficult to determine which of these differences is statistically significant. A multivariable logistic regression will be difficult given that the outcome has three categories, but some idea of how these characteristics vary, or confound each other, may be useful.

In this paper the intention of Table 1 is simply to describe the characteristics in the 3 groups rather than to provide a detailed investigation of how these characteristics vary/confound each other. Our regression model focusing on the associates of interest are adjusted for factors identified as important potential confounders based on causal inference models and external information, i.e., not data driven.

3. Residual confounding by cancer type – Though the sample size is likely insufficient to categorize cancer type any more finely, it would be important to acknowledge as a limitation that residual confounding likely exists within the current classification. AML vs. ALL, or HL vs. NHL for example, may well have different findings from each other when looking at difference between treatment setting, that would be obscured.

Thank you for your comment, it is a bit unclear; we have read it as being about how we adjust for cancer type as a confounder rather than considering effects by subgroup (which is what you are referring to). We adjusted for cancer type as a confounder in our models using categories: leukaemia, lymphoma, brain and central nervous system, bone tumours, sarcoma, germ cell, melanoma, carcinomas, other. We were not able to use finer categories because of small numbers.

4. Discussion – I feel the Discussion underplays the significance of the non survival outcomes being different by treatment care setting. There is a rich literature on the importance of psychosocial care and fertility discussions to AYA, so even if the finding of no survival differences is true, this alone may be a point that demonstrated the value of these TYA centres. This warrants much more discussion than it currently receives in the manuscript.

Thank you for your comment, we have added this to conclusions which only mentioned the survival outcomes. 'Young people were more likely to have had documentation of access to supportive care services or have a fertility discussion if they had some or all of their care delivered by the TYA-PTC, which existing literature supports as important for TYA.'

We also feel this paragraph in the discussion answers your query.

'It is therefore disappointing that there were no survival differences noted between the three categories. Alternatively, we could view this as a positive finding - wherever young people choose to be treated, their survival outcomes are the same. The work we did with young people to develop BRIGHTLIGHT highlighted that they did not perceive survival alone as the most important outcome. Quality of life and the ability to get back on with life were as important (18,19). While we have shown a better improvement in quality of life when treated in a TYA-PTC (23), we have yet to ascertain whether young people's reintegration into life when treatment ends is also better.'

5. The finding of possible differences in survival among older BRIGHTLIGHT AYA is intriguing. The authors note that this group (>19 years) is theoretically given a choice as to where their treatment is received. Another factor which perhaps should be discussed is that the alternative for younger TYA to TYA-PTC will likely be paediatric cancer centres, while that for ≥ 19 yo will be adult centres. Many studies have shown that even in systems without AYA specific units/centres, care differs between these two treatment settings for AYA. Is it possible to perform a sensitivity analysis where patients in the NONE or SOME categories are excluded if their non PTC care was received in a paediatric centre? I realize that this may represent additional work for the authors, but may be a very important finding.

While this is an interesting observation, we used Hospital Episode Statistics admitted patient care data to determine where young people were treated and this is at hospital level data, not ward or departmental level. There are only a small number of children's hospitals in England and not all of these were open to recruitment (as they stated they did not treat the eligible age group) so the numbers from these hospitals will be prohibitively small. Only 56 respondents self-identified that a

children's ward was their main place of care; again the number is too small to undertake any sensitivity analysis.

MINOR COMMENTS

Table 3 – Consider renaming the ALL category to avoid confusion between ALL and ALL-TYA-PTC.

We have changed the heading to 'TOTAL' to avoid confusion.